# Infection prevention and control risk factors in health workers infected with SARS-CoV-2 in Jordan: A case control study

**Ala Bin Tarif**[1], **Mohannad Ramadan**[2], **Mo Yin**[3,4,5,6,7], **Ghazi Sharkas**[1], **Sami Sheikh Ali**[1], **Mahmoud Gazo**[1], **Ali Zeitawy**[1], **Lora Alsawalha**[2], **Kaiyue Wu**[8], **Alvaro Alonso-Garbayo**[2], **Bassim Zayed**[8], **Lubna Al-Ariqi**[8], **Khalid A. Kheirallah**[9], **Maha Talaat**[8], **Arash Rashidian**[8], **Alice Simniceanu**[3], **Benedetta Allegranzi**[3], **Alessandro Cassini**[3], **Saverio Bellizzi**[2]*

1 Jordan Ministry of Health, Amman, Jordan, 2 WHO Jordan Country Office, Amman, Jordan, 3 WHO Headquarters, Geneva, Switzerland, 4 Centre for Tropical Medicine and Global Health, Nuffield Department of Medicine, University of Oxford, Oxford, United Kingdom, 5 Mahidol-Oxford Tropical Medicine Research Unit, Faculty of Tropical Medicine, Mahidol University, Bangkok, Thailand, 6 Division of Infectious Diseases, University Medicine Cluster, National University Hospital, Singapore, Singapore, 7 Department of Medicine, National University of Singapore, Singapore, Singapore, 8 WHO Regional Office for the Eastern Mediterranean, Cairo, Egypt, 9 Department of Public Health, Faculty of Medicine, Jordan University of Science and Technology, Irbid, Jordan

* bellizzis@who.int

**Data Availability Statement:** Proprietors of the data are the Jordanian Ministry of Health, which can make such data available upon request. The authors of this study can facilitate access to the

## Abstract

### Background

Despite under-reporting, health workers (HWs) accounted for 2 to 30% of the reported COVID-19 cases worldwide. In line with data from other countries, Jordan recorded multiple case surges among HWs.

### Methods

Based on the standardized WHO UNITY case-control study protocol on assessing risk factors for SARS-CoV-2 infection in HWs, HWs with confirmed COVID-19 were recruited as cases from eight hospitals in Jordan. HWs exposed to COVID-19 patients in the same setting but without infection were recruited as controls. The study lasted approximately two months (from early January to early March 2021). Regression models were used to analyse exposure risk factors for SARS-CoV-2 infection in HWs; conditional logistic regressions were utilized to estimate odds ratios (ORs) adjusted for the confounding variables.

### Results

A total of 358 (102 cases and 256 controls) participants were included in the analysis. The multivariate analysis showed that being exposed to COVID-19 patients within 1 metre for more than 15 minutes increased three-fold the odds of infection (OR 2.92, 95% CI 1.25–6.86). Following IPC standard precautions when in contact with patients was a significant protective factor. The multivariate analysis showed that suboptimal adherence to hand hygiene increased the odds of infection by three times (OR 3.18; 95% CI 1.25–8.08).

dataset through the Jordanian Ministry of Health. The researchers cannot make data available directly on behalf of the Jordanian Ministry of Health. Retrieval of data should be sought via the Jordanian Ministry of Health (iprd@moh.gov.jo).

**Funding:** This work has been supported by German Federal Ministry of Health (BMG) COVID-19 Research and development funding to the World Health Organization. The funders had no role in study design, data collection and analysis, decision to publish, or preparation of the manuscript.

**Competing interests:** The authors have declared that no competing interests exist.

## Conclusion

Study findings confirmed the role of hand hygiene as one of the most cost-effective measures to combat the spreading of viral infections. Future studies based on the same protocol will enable additional interpretations and confirmation of the Jordan experience.

## Introduction

The current COVID-19 pandemic has accounted for almost 250 million people infected and 5 million deaths worldwide as of November 2021 [1], disproportionally affecting health workers (HWs). Despite possible under-reporting, HWs accounted for 2 to 30% of the reported COVID-19 cases worldwide [2].

Only 6643 deaths out of the 3.45 million deaths due to COVID-19 between January 2020 and May 2021 were identified as being in HWs; however, this figure significantly under-reports the burden of mortality world-wide in this group [3].

The WHO defines HWs as "all people engaged in actions whose primary intent is to enhance health" [4]. HWs play a critical role in providing patient care and preventing further transmission. Hence, insights on the risk factors for SARS-CoV-2 infections among HWs are needed to protect HWs and mitigate the risk of onward transmission of infection.

Jordan reported almost 805 000 confirmed COVID-19 cases and 10 500 deaths by the beginning of September 2021 [5]. These represented about 5.0% of the total confirmed cases and 4.0% of the total number of deaths in the WHO Eastern Mediterranean Region (EMR) [6]. The COVID-19 epidemiological curve in Jordan during the first 19 months of the pandemic showed four distinct phases that reflected: 1. the complex interrelation between the natural evolution of the outbreak, 2. the implementation of public health and social measures (PHSM), 3. the introduction of variants of concern (VoC) 4. and the COVID-19 vaccination campaign. The first phase started last week of February 2020 when the government applied strict control measures, thus flattening the epidemiological curve and prolonging sporadic transmission [7]. The second phase featured progressive easing of restrictions with an exponential increase of cases up to 8 000 in November 2020 [7]. A third phase showed a new steady and progressive upsurge of cases with a peak of almost 10 000 cases per day over the last week of March 2021 (most likely due to the introduction of the *Alpha* VoC) [5]. The fourth phase showed a steady and progressive decline of the epidemiological curve with a long plateau of around 900 cases per day during the June-September 2021 time-period [7]. To note that the COVID-19 vaccination campaign in Jordan started in mid-January 2021 and targeted all individuals regardless of nationality, citizenship, and legal status [8].

During the first peak of COVID-19 pandemic in Jordan (November 2020), a total of 817 cases were recorded among nurses, representing 5.5% of HWs, and 26 deaths were recorded among physicians [9].

To quantify SARS-CoV-2 exposure risks for HWs and identify effective protective measures, WHO developed a protocol for a case-control study *"Assessment of risk factors for coronavirus disease 2019 (COVID-19) in health workers: protocol for a case-control study"* and engaged countries in a global multi-centre study [10].

Jordan was one of the participating countries. Here we report the findings from the Jordan study, which lasted approximately two months (from early January to early March 2021). and provide recommendations to improve IPC measures in healthcare facilities across the country within the context of the response to the COVID-19 pandemic.

## Methods

According to study enrolment criteria and interest, eight hospitals with an approximate pooled number of 6,000 health workers were selected to participate in the study.

### Study design and participant enrolment

A HW was defined as any staff in the health care facility involved in the provision of patient care, including health care professionals, allied health workers and auxiliary health workers such as cleaning and laundry personnel, x-ray physicians and technicians, clerks, phlebotomists, respiratory therapists, nutritionists, social workers, physical therapists, laboratory personnel, cleaners, admission/reception clerks, patient transporters, and catering staff. Exposure to COVID-19 patients was defined as close contact (within 1 metre and for more than 15 minutes) with a suspected/probable/confirmed COVID-19 patient(s), or indirect contact with fomites (for example, clothes, linen, utensils, furniture and so on) or with materials, devices or equipment linked to a suspected/probable/confirmed COVID-19 patient(s).

### Identification of cases

A case was defined as a health worker exposed in a health care setting to a COVID-19 patient in the 14 days prior to the health worker's confirmation test, and who is a confirmed COVID-19 case fulfilling either of the three criteria below:

- A person with a positive Nucleic Acid Amplification Test (NAAT)

- A person with a positive SARS-CoV-2 Ag-RDT AND meeting either the probable case definition or suspected criteria A OR B

- An asymptomatic person with a positive SARS-CoV-2 Ag-RDT AND who is a contact of a probable or confirmed case.

HWs were excluded if they were vaccinated more than 2 weeks prior to their first positive confirmation test, or if they were a close contact of a COVID-19 case outside of work.

### Selection of controls

HWs exposed to COVID-19 patients in the same setting as the cases in the 14 days prior to enrolment, without suspected, confirmed or previous infection with SARS-CoV-2 infections were recruited as controls, with a target of at least 2–4 controls for every case. The exclusion criteria for a control were having a positive serology test to SARS-CoV-2 and/or prior vaccination.

### Data

Data collection and entry was performed on Go.Data [11]. To ensure data quality across the various sites, data entered was checked for accuracy, timeliness, and completeness prior to merging and analysis. Data variables obtained from the questionnaires (**Fig 1**) include the following broad categories:

- Demographic factors, e.g., age, sex, country of residence, educational level;

- Personal risk factors, e.g., occupation, hygiene practices, various types of exposures to SARS-CoV-2;

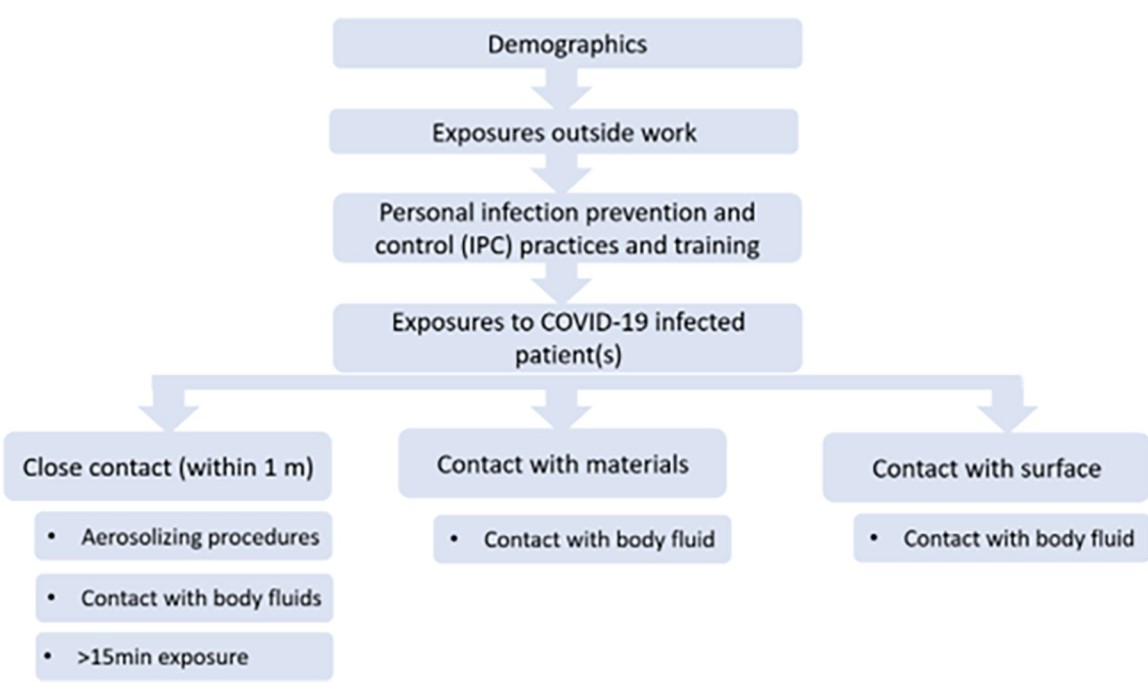

**Fig 1. Questionnaire structure.**

- Risk factors related to healthcare facilities, e.g., IPC policies, available personal protective equipment (PPE) resources; and,

- Outcomes, e.g., infection with SARS-CoV-2, mortality, hospitalisation, serological response.

## Statistical analysis

Descriptive statistics include frequency tables for categorical data, means (with standard deviations), or medians (with interquartile ranges) depending on the distribution of the data. Categorical variables were compared with $\chi 2$ and Fisher exact tests as appropriate and continuous variables with unpaired, 2-tailed t tests or nonparametric Wilcoxon rank sum tests as appropriate.

We used conditional logistic regressions to estimate odds ratios (ORs) adjusted for the confounding variables. Because the cases and controls were matched based on the health care facilities when enrolled into the study, health care facility was used as grouping strata (matching variable) in the conditional regression model. Collinearity of the infection risk factors were evaluated in separate multivariate models adjusted for age, sex, occupation, education level, and country of residence with Cramer's V measure.

The following models were considered:

- Model 1: Associations between demographic factors and IPC practices and SAR-CoV-2 infection.

- Model 2: Associations between individual IPC practices and COVID-19 infection.

- Model 3: Associations between exposure-specific protection and risk factors and SARS--CoV-2 infection during high-risk exposures.

These high-risk procedures include:

○ Close contact (within 1 metre) with COVID-19 patient(s)

○ During exposure to COVID-19 patient(s)' materials

○ During exposure to COVID-19 patient(s)' surrounding surfaces

Planned variables in the regression models were categorised and chosen based on expert opinion from IPC specialists together with statisticians. Confounders including demographics, community exposure to SARS-CoV-2, occupational roles (and IPC training in the models evaluating PPE items and infection risk) were included in all the models. Actual variables included in the models were finalised using variables which had adequate variations in the responses and the responses did not correlate highly with other variables. Univariate regressions were performed prior to multivariate regressions. Final multivariate regression models were selected based on Akaike information criterion (AIC) values.

### Ethical approval

Institutional Review Board (IRB) approval was obtained from the Jordan Ministry of Health Ethical Committee (Reference # 2021/MBA/14375).

### Results

A total of 358 (102 cases and 256 controls) individuals were enrolled into the study between 8 January 2021 and 4 March 2021 across the eight participating centres. The mean time interval between the first and follow-up interviews was 23 days (IQR 21 to 26 days).

As per **Table 1**, six out of ten participants in the study were female (N = 210), which contributed to slightly more than half of cases and 60% of controls. Almost 90% of participants held a tertiary or university education level, while only one out of 10 participants in the study were medical doctors. In terms of occupations, more participants with little patient contact e.g. administration clerks were enrolled as controls. However, in the multivariate regression (**Table 2**), occupational role was not an important factor associated with SARS-CoV-2 infection. On the other hand, close contact and duration of exposure to a COVID-19 patient increased three-fold the odds of acquiring infection (OR 3.13; 95% CI 1.71–5.70).

Slightly less than 20% of both cases and controls reported exposure to body fluids and aerosol procedures, which did not lead to a significant increased risk (exposure to body fluid: OR 0.49; 95% CI 0.17–1.43; exposure to aerosol: OR 1.61 95% CI 0.50–1.17). Conversely, not always practicing the five-moments of hand hygiene [12] was related to more than 40% of cases (N = 41) and around 25% of controls (N = 65), which yielded an OR of 3.18 (95% CI 1.25–8.08) in multivariate regression. Almost nine out of ten participants reported that PPE was available at their respective health facility; "COVID-19 specific IPC training" as well as "hours of training" were equally distributed across cases and controls. Having received training remotely seemed to have decreased the odds of infection when compared to the baseline category (OR <0.05; 95% CI 0.04–0.40), which grouped those who benefitted from both practical and theoretical training.

The second statistical model was based on 51 cases and 83 controls, which represented the sample of participants that participated in high-risk exposure procedures. Both hand-hygiene before and after any procedure during close contact reduced significantly the odds of infection (**Table 2**). On the other hand, the analysis assessing the role of exposure to body fluids and aerosol generating procedures yielded statistically non-significant results.

The last statistical model was based on an even lower sample of participants (31 cases and 26 controls) who had prolonged close contact (>15 min within 1 meter), which did not allow

**Table 1. Associations between demographic factors and IPC practices and COVID-19 infection.**

| Characteristic | Cases | Controls | Univariate Analysis | | | Multivariate Analysis** | | |
|---|---|---|---|---|---|---|---|---|
| | | | OR | 95% CI | p-value | aOR | 95% CI | p-value |
| Gender | | | | | | | | |
| Female | 54 (52.9%) | 156 (60.9%) | - | - | - | - | - | - |
| Male | 48 (47.1%) | 100 (39.1%) | 1.38 | 0.86–2.21 | 0.18 | 1.83 | 0.90–3.73 | 0.10 |
| Education | | | | | | | | |
| Tertiary/University | 92 (90.2%) | 216 (84.4%) | - | - | - | - | - | - |
| Secondary and below | 10 (9.8%) | 40 (15.6%) | 0.5 | 0.24–1.06 | 0.07 | 0.83 | 0.29–2.40 | 0.72 |
| Health worker role | | | | | | | | |
| Medical doctor | 16 (15.7%) | 22 (8.6%) | - | - | - | - | - | - |
| Nurse | 46 (45.1%) | 89 (34.8%) | 0.88 | 0.41–1.87 | 0.73 | 0.88 | 0.27–2.83 | 0.83 |
| Other* | 40 (39.2%) | 145 (56.6%) | 0.40 | 0.19–0.85 | 0.02 | 0.74 | 0.22–2.45 | 0.60 |
| Providing COVID-19 specific care | | | | | | | | |
| Yes | 26 (25.5%) | 47 (18.4%) | - | - | - | - | - | - |
| No | 76 (74.5%) | 209 (81.6%) | 0.61 | 0.35–1.06 | 0.08 | 1.10 | 0.39–3.11 | 0.9 |
| Exposed to COVID-19 patients within 1 meter distance for more than 15 min | | | | | | | | |
| No | 57 (55.9%) | 200 (78.1%) | - | - | - | - | - | - |
| Yes | 31 (30.4%) | 36 (14.1%) | 3.13 | 1.71–5.70 | <0.05 | 2.92 | 1.25–6.86 | 0.01 |
| Missing | 14 (13.7%) | 20 (7.8%) | - | - | - | - | - | - |
| Exposed to COVID-19 patients within 1 meter distance with aerosol procedure | | | | | | | | |
| No | 73 (71.6%) | 204 (79.7%) | - | - | - | - | - | - |
| Yes | 15 (14.7%) | 32 (12.5%) | 1.49 | 0.75–2.96 | 0.26 | 1.61 | 0.50–5.17 | 0.45 |
| Missing | 14 (13.7%) | 20 (7.8%) | - | - | - | - | - | - |
| Exposed to surfaces around COVID-19 patients soiled with body fluid | | | | | | | | |
| No | 72 (70.6%) | 204 (79.7%) | - | - | - | - | - | - |
| Yes | 18 (17.6%) | 41 (16.0%) | 1.49 | 0.78–2.84 | 0.22 | 0.49 | 0.17–1.43 | 0.22 |
| Missing | 12 (11.8%) | 11 (4.3%) | - | - | - | - | - | - |
| Aware of the hand hygiene five moments | | | | | | | | |
| No | 34 (33.3%) | 113 (44.1%) | - | - | - | - | - | - |
| Yes | 68 (66.7%) | 143 (55.9%) | 2.5 | 1.41–4.41 | <0.05 | 1.99 | 0.87–4.53 | 0.10 |
| Hand hygiene five moments practice | | | | | | | | |
| Always | 61 (59.8%) | 191 (74.6%) | - | - | - | - | - | - |
| Not always | 41 (40.2%) | 65 (25.4%) | 1.76 | 1.03–3.03 | 0.04 | 3.18 | 1.25–8.08 | 0.01 |
| Personal protective equipment available in the facility | | | | | | | | |
| Yes | 85 (83.3%) | 232 (90.6%) | - | - | - | - | - | - |
| No | 17 (16.7%) | 24 (9.4%) | 1.2 | 0.89–3.70 | 0.55 | 1.44 | 0.48–4.34 | 0.58 |
| Received IPC training specific to COVID-19 | | | | | | | | |
| No | 56 (54.9%) | 193 (75.4%) | - | - | - | - | - | - |
| Yes | 46 (45.1%) | 63 (24.6%) | 2.53 | 1.51–4.24 | <0.05 | 1.51 | 0.62–3.65 | 0.41 |
| Received IPC training specific to COVID-19 in person | | | | | | | | |
| Both | 40 (39.2%) | 31 (12.1%) | - | - | - | - | - | - |
| Only practical | 23 (22.5%) | 27 (10.5%) | 0.97 | 0.44–2.14 | 0.95 | 1.36 | 0.47–3.95 | 0.6 |
| Remotely/Theoretical | 23 (22.5%) | 133 (52.0%) | 0.1 | 0.05–0.22 | <0.05 | 0.13 | 0.04–0.40 | <0.05 |
| Don't know IPC standard | 16 (15.7%) | 65 (25.4%) | 0.16 | 0.07–0.35 | <0.05 | 0.23 | 0.05–1.05 | 0.06 |
| Hours of IPC training | | | | | | | | |
| More than two hours | 44 (43.1%) | 42 (16.4%) | - | - | - | - | - | - |

(*Continued*)

**Table 1.** (Continued)

| Characteristic | Cases | Controls | Univariate Analysis | | | Multivariate Analysis** | | |
|---|---|---|---|---|---|---|---|---|
| | | | OR | 95% CI | p-value | aOR | 95% CI | p-value |
| Less than two hours | 58 (56.9%) | 214 (83.6%) | 0.25 | 0.14–0.42 | <0.05 | 0.59 | 0.23–1.50 | 0.33 |

*Other included: laboratory personnel, admission and reception clerk, cleaner, radiology technician, patient transporter, phlebotomist, physical therapist, and catering staff

**Confounders included contact covid outside of work, used public transport, had social contact outside of work

for the multivariate logistic regression to be conducted. There was also inadequate power in the univariate analysis to find any associations between a PPE item and infection (Table 3).

## Discussion

The HWs case-control study in Jordan is the first documented experience within the WHO multi-centre COVID-19 WHO UNITY case-control study global initiative, which aimed to identify specific IPC risk factors for SARS-CoV-2 infection in health workers. The pooled analysis using data from eight hospitals revealed that not practicing HH according to minimum standards was associated with more than three-fold odds of being infected with SARS-CoV-2.

WHO has declared 2021 the "Year of the Health and Care Worker" and evidence has shown that appropriate hand hygiene practices are vital to protect such vital workers by reducing infections during care delivery [13]. However, very few studies have so far explored the effect of hand hygiene on COVID-19 among health workers and only one detected significant association, although based on a systematic review of the literature [14]. Our finding suggested that hand hygiene both before and after procedures (respectively, OR 0.23, 95% CI 0.04–0.88, and OR 0.24, 95% CI 0.06–0.99) are in line with what reported by Ran et al., who detected that unqualified hand hygiene led to a 2.64 risk ratio (RR) of developing infection; similarly, suboptimal hand hygiene before patient contact and suboptimal hand hygiene after patient contact were respectively associated with a 3.10 RR (1.43–6.73) and 2.43 RR (1.34–4.39) [15]. A recent

**Table 2. Associations between individual IPC practices during contact within 1 metre and SARS-CoV-2 infection.**

| Characteristic | Cases | Controls | Univariate Analysis | | | Multivariate Analysis* | | |
|---|---|---|---|---|---|---|---|---|
| | | | OR | 95% CI | p-value | aOR | 95% CI | p-value |
| Hand hygiene before providing care | | | | | | | | |
| Not always | 19 (37.3%) | 11 (13.3%) | - | - | - | - | - | - |
| As recommended | 32 (62.7%) | 72 (86.7%) | 0.25 | 0.08–0.75 | 0.01 | 0.23 | 0.04–0.88 | <0.05 |
| Hand hygiene after providing care | | | | | | | | |
| No | 16 (31.4%) | 10 (12.0%) | - | - | - | - | - | - |
| Yes | 35 (68.6%) | 73 (88.0%) | 0.25 | 0.07–0.82 | 0.02 | 0.25 | 0.06–0.99 | 0.05 |
| During aerosol-generating procedure | | | | | | | | |
| No | 36 (70.6%) | 51 (61.4%) | - | - | - | - | - | - |
| Yes | 15 (29.4%) | 32 (38.6%) | 0.69 | 0.30–1.57 | 0.37 | 0.65 | 0.26–2.24 | 0.76 |
| During exposure to surfaces soiled with body fluid | | | | | | | | |
| No | 35 (68.6%) | 52 (62.7%) | - | - | - | - | - | - |
| Yes | 14 (27.5%) | 30 (36.1%) | 0.81 | 0.33–1.96 | 0.63 | 0.63 | 0.21–1.87 | 0.42 |
| Missing | 2 (3.9%) | 1 (1.2%) | - | - | - | - | - | - |

*Confounders included awareness of hand hygiene moments, received IPC training specific to COVID-19, contact covid outside of work, used public transport

**Table 3. Associations between PPE during prolonged close contact and COVID-19 infection.**

| Characteristic | Cases | Controls | Univariate Analysis | | | Multivariate Analysis* | | |
|---|---|---|---|---|---|---|---|---|
| | | | OR | 95% CI | p-value | aOR | 95% CI | p-value |
| face shield | | | | | | | | |
| Yes | 16 (51.6%) | 20 (55.6%) | - | - | - | - | - | - |
| No | 15 (48.4%) | 16 (44.4%) | 1.14 | 0.33–3.99 | 0.84 | - | - | - |
| gloves | | | | | | | | |
| Yes | 19 (61.3%) | 23 (63.9%) | - | - | - | - | - | - |
| No | 12 (38.7%) | 13 (36.1%) | 0.96 | 0.30–3.06 | 0.94 | - | - | - |
| coverall | | | | | | | | |
| Yes | 22 (71.0%) | 29 (80.6%) | - | - | - | - | - | - |
| No | 9 (29.0%) | 7 (19.4%) | 1.61 | 0.44–5.83 | 0.91 | - | - | - |
| head cover | | | | | | | | |
| Yes | 17 (54.8%) | 18 (50.0%) | - | - | - | - | - | - |
| No | 14 (45.2%) | 18 (50.0%) | 0.56 | 0.18–1.77 | 0.32 | - | - | - |
| respirator (e.g. N95, FFP2 or equivalent) | | | | | | | | |
| Yes | 13 (41.9%) | 13 (36.1%) | - | - | - | - | - | - |
| No | 18 (58.1%) | 23 (63.9%) | 0.52 | 0.14–1.94 | 0.33 | - | - | - |
| shoe cover | | | | | | | | |
| Yes | 9 (29.0%) | 12 (33.3%) | - | - | - | - | - | - |
| No | 22 (71.0%) | 24 (66.7%) | 1.07 | 0.31–3.73 | 0.91 | - | - | - |
| medical/surgical mask | | | | | | | | |
| Yes | 13 (41.9%) | 11 (30.6%) | - | - | - | - | - | - |
| No | 18 (58.1%) | 25 (69.4%) | 0.29 | 0.08–1.09 | 0.07 | - | - | - |
| goggles | | | | | | | | |
| Yes | 13 (41.9%) | 12 (33.3%) | - | - | - | - | - | - |
| No | 18 (58.1%) | 24 (66.7%) | 0.31 | 0.07–1.24 | 0.09 | - | - | - |
| gown | | | | | | | | |
| Yes | 17 (54.8%) | 14 (38.9%) | - | - | - | - | - | - |
| No | 14 (45.2%) | 22 (61.1%) | 0.36 | 0.22–1.08 | 0.76 | - | - | - |

*Confounders included awareness of hand hygiene moments, received IPC training specific to COVID-19, contact covid outside of work, used public transport

study in Jordan indicated sub-optimal precautionary behaviour of medical doctors in context of COVID-19 in early pandemic and implicitly revealed that only 47.4% of doctors were practicing proper hand hygiene during duty hours. These results present a piece of evidence calling for more actions to boost preventive behaviour among HWs [16].

Other reports from the WHO Eastern Mediterranean Region indicated mixed results on the role of hand hygiene towards COVID-19. A cross-sectional study conducted in Egypt showed no significant protective role by proper hand hygiene (OR 0.74; 95% CI 0.15–3.59) [17]; the same study found no significant results when assessing the use of PPE as recommended (OR 1.00; 95% CI 0.21–4.72) [17]. Another analysis from Egypt revealed very similar findings with no statistically significant protective role of proper hand hygiene (OR 0.26; 95% CI 0.02–4.46) [18]. As far as reports from other WHO regions are concerned, a case-control study from Bangladesh yielded no conclusive results on the role of hand hygiene in the different phases: during patient care (OR 0.78, 95% CI 0.23–2.67); during procedure (OR 3.28, 95% CI0.66–12.30); after body fluid exposure (OR 0.28, 95% CI 0.06–1.45); after touching fomites (OR 1.58, 0.49–5.04) [19].

The risk of SARS-CoV-2 transmission in health facilities may increase in relation to aspects such as lack of adequate isolation facilities and increased demand for hospital beds for patients [14], as well the lack of PPE availability and the need to conduct high-risk procedures such as aerosol generating procedures [13]. Also, to note that health workers are members of communities and as such can play a role in transmission between health-care settings and the community, which might have a key role in amplifying outbreaks in settings such as health facilities [20]. However, most evidence around exposure determinants for infection remains low or moderate certainty because of methodological limitations, imprecision, and inconsistency [14].

Unlike the available literature [14], we did not detect a significant higher proportion of exposure to COVID-19 patients as well as exposure to contact with bodily secretions and aerosol generating procedures among cases. While physical distancing of at least 1 metre remains a key IPC and public health and social measure to reduce transmission of SARS-CoV-2 and has been substantiated in several analyses, the available evidence mostly stems from studies with small sample sizes and other methodological concerns [21–23].

In our univariate analysis, the proportional distribution of having received IPC training and correctly applying IPC standards was significantly higher among controls. However, such an association lost its statistical significance in the adjusted analysis. Most of previous studies on IPC training mostly focused on personal protective equipment (PPE) use. An Italian cross-sectional study in 2020 found no statistical difference between PPE training vs. no training [24], and very similar findings were found in another cross-sectional study based in the US; however, both studies might be affected by recall bias [1]. Interestingly, our multivariate analysis showed a protective effect of IPC training lasting less than 2 hours when compared with longer trainings (OR 0.46; 95% CI 0.22–0.98); such a result may not be easy to interpret and could be linked to targeted training for infection prevention and control, such as specific transmission prevention measures that HW could implement easily.

For the variable on the "use of PPE when indicated", our adjusted analysis did not detect any significant difference between cases and controls. Available literature provides mixed findings on this subject while one recent study found that consistent mask use was associated with a lower risk of infection when compared to inconsistent use [25], another study found no significant decrease while wearing PPE and exposed to a COVID-19 infected patient [26]. To note how non-significant findings on this association are likely linked to the cross-sectional nature of several studies, which does not allow for proper ascertainment of infection timing relative to different exposures.

Our findings further complement the role of hand hygiene as an essential component of IPC, which is often neglected by HWs both in developed and developing countries, with compliance rates sometimes dipping below 20% [27]. Reasons for this include aspects like overcrowding of healthcare facilities and absence of reliable and adequate hand cleaning infrastructure, such as clean water and hand washing stations, or alcohol-based hand rub (ABHR) hand hygiene dispensers at the point of care especially in low-resourced settings [28]. These factors may explain why some studies report rates as low as 1 in 10 health workers practice proper hand hygiene while caring for patients at high risk of health care-associated infections in high-risk settings such as the ICU [29]. This translates into patients in low- and middle-income countries to be twice as likely to experience high rates of hospital acquired infections during health care delivery when compared to patients in high-income countries (15% and 7% of patients respectively) [13].

There are some limitations with this study that were due to the evolving nature of the pandemic and the writing of the protocol which was done in early 2020. Changes to the epidemiology of health worker infections are driven by transmission patterns: a first phase with high

from transmission in health care facilities; a second phase featuring prominent community transmission which may have contributed to both healthcare transmission and outward from the healthcare facility to the community. Thus, the acquisition of SARS-CoV-2 infection in the community is not well determined by this study and therefore may compound some findings. Furthermore, due to the nature of the study, we cannot underestimate the role of recall bias. However, we deem it to be non-differential as the time between the RT-PCR results and the interview were similar between groups. Our results on the importance of hand washing and for training and lower importance of PPE is in contrast with findings from other published reports; this might be caused by bias like in the case of those who are rigorous hand washers more likely to be those with highest adherence to other protective measures. As fare as selection bias is concerned, reasons for declining participation were similar between groups and were mainly related to availability, which makes selection bias unlikely. Additional confounding could be present due to unmeasured variables such as the prevalence of the infection in the place of residence as well as the quality of training. We must acknowledge that the study sample was not random and included consecutive subjects in a defined time frame. Finally, generalization is not possible due to important differences in the pandemic evolving differently across countries, regions, and hospitals. Specifically, during the study period health workers were most likely affected by the original coronavirus strain, which was less contagious than the subsequent VOCs.

Findings from this study reinforce the infection prevention and control measures that WHO has highlighted during the current pandemic [30], such as the importance of hand hygiene, appropriate PPE wear, distancing from patients when not providing care and providing regular IPC training to health workers. The results of this analysis were consistently utilized to tailor infection and prevention control capacity building Jordan MoH-WHO joint activities at all health care facilities. Future findings from the larger WHO multi-centre study will enable additional interpretations and confirmation of the experience in Jordan.

## Author Contributions

**Conceptualization:** Ala Bin Tarif, Alvaro Alonso-Garbayo, Maha Talaat, Arash Rashidian, Alice Simniceanu, Benedetta Allegranzi, Alessandro Cassini, Saverio Bellizzi.

**Data curation:** Mohannad Ramadan, Mo Yin, Sami Sheikh Ali, Lora Alsawalha, Kaiyue Wu, Khalid A. Kheirallah, Alessandro Cassini.

**Formal analysis:** Mohannad Ramadan, Mo Yin, Sami Sheikh Ali.

**Funding acquisition:** Maha Talaat, Arash Rashidian, Alice Simniceanu, Benedetta Allegranzi.

**Investigation:** Ala Bin Tarif, Mohannad Ramadan, Mo Yin, Ghazi Sharkas, Sami Sheikh Ali, Lora Alsawalha, Kaiyue Wu, Bassim Zayed, Benedetta Allegranzi, Alessandro Cassini.

**Methodology:** Mo Yin, Ghazi Sharkas, Lora Alsawalha, Kaiyue Wu, Bassim Zayed, Lubna Al-Ariqi, Maha Talaat, Arash Rashidian, Alice Simniceanu, Benedetta Allegranzi, Alessandro Cassini.

**Project administration:** Ala Bin Tarif, Mahmoud Gazo, Ali Zeitawy, Lora Alsawalha, Lubna Al-Ariqi, Maha Talaat, Benedetta Allegranzi, Alessandro Cassini.

**Resources:** Ghazi Sharkas, Mahmoud Gazo, Ali Zeitawy, Bassim Zayed, Lubna Al-Ariqi, Khalid A. Kheirallah, Maha Talaat, Arash Rashidian, Alice Simniceanu, Benedetta Allegranzi, Alessandro Cassini.

**Software:** Ala Bin Tarif, Ghazi Sharkas, Mahmoud Gazo, Ali Zeitawy, Bassim Zayed, Alessandro Cassini.

**Supervision:** Ala Bin Tarif, Sami Sheikh Ali.

**Validation:** Mohannad Ramadan, Mo Yin, Ghazi Sharkas, Mahmoud Gazo, Lora Alsawalha, Kaiyue Wu, Khalid A. Kheirallah, Alice Simniceanu, Benedetta Allegranzi, Saverio Bellizzi.

**Visualization:** Mo Yin, Ghazi Sharkas, Sami Sheikh Ali, Kaiyue Wu, Alvaro Alonso-Garbayo, Lubna Al-Ariqi, Khalid A. Kheirallah, Arash Rashidian.

**Writing – original draft:** Ala Bin Tarif, Mo Yin, Sami Sheikh Ali, Ali Zeitawy, Lora Alsawalha, Alvaro Alonso-Garbayo, Bassim Zayed, Maha Talaat, Alice Simniceanu, Benedetta Allegranzi, Alessandro Cassini, Saverio Bellizzi.

**Writing – review & editing:** Ala Bin Tarif, Mohannad Ramadan, Ghazi Sharkas, Mahmoud Gazo, Ali Zeitawy, Lora Alsawalha, Kaiyue Wu, Alvaro Alonso-Garbayo, Bassim Zayed, Lubna Al-Ariqi, Khalid A. Kheirallah, Maha Talaat, Arash Rashidian, Alice Simniceanu, Benedetta Allegranzi, Alessandro Cassini, Saverio Bellizzi.

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
