## [Decision Letter · Decision Letter 0]

13 Apr 2022

PONE-D-22-04948Infection prevention and control risk factors in health workers infected with SARS-CoV-2 in Jordan: a case control studyPLOS ONE

Dear Dr. Bellizzi,

Thank you for submitting your manuscript to PLOS ONE. After careful consideration, we feel that it has merit but does not fully meet PLOS ONE’s publication criteria as it currently stands. Therefore, we invite you to submit a revised version of the manuscript that addresses the points raised during the review process.

Please see the reviewer comments attached.

In addition to those, please specify the timeframe of your study - this will help understand which variant was dominant at the time.

Please also consider including any information on the impact of different VOCs/VOIs in the studied samples.

Reviewer #1: Thanks for the authors for the good work, comments as below:

1- In introduction section line -75 & discussion section line-318, It would be great if the COVID-19 epidemiological curve in Jordan during the first 19 months of the pandemic can be demonstrated in a figure that reflect count of confirmed covid-19 cases in community and counts of infected HWs in 8 hospitals in parallel with actions taken at the public & hospital level. See an example below:

https://doi.org/10.1001/jama.2020.11160

2- Please indicate what is the status of 8 hospitals in regard to similarities in IPC practices, PPE shortages and designated COVID-19 hospitals.

3- Under study design & participants enrollment:

Line -115, please define a suspected/probable COVID-19 patients or indicate the references in reference section.

4- Under identification of cases subsection: RDT abbreviation stands for what- need to write it as is in the first time.? Also, what do you mean by suspected criteria A or B in line 124 (please clarify).

5- Under control section: How did you manage HWs with prior positive PCR or RDT , did you test them before enrollment especially they can carry the virus with no symptoms, how can you handle it.

Reviewer #2: This study supports the COVID-19 prevention measures adopted in hospitals located in Jordan as advised by the WHO. The results highlights the importance of personal hygiene in combating viral infections as well as in precautions to pandemic. The study is short but has impact in the current situation.

Minor comments

- The standard method of hand washing needs to be mentioned with other PPEs. A short summary of the training provided to HWs needs to be mentioned.

- Whether the HWs were trained for hand washing steps, especially for this study?

- The HWs were wearing gloves as part of their PPEs. How does the investigators implement hand washing in this context? Do they change the gloves very often during their work?

- Authors stated that HWs exposed to COVID-19 patients within 1 meter for more than 15 minutes increased 3-fold adds of infection. As SARS-CoV-2 is a respiratory virus, a more detailed clarification with respect to the face mask need to be described.

We look forward to receiving your revised manuscript.

Kind regards,

Balram Rathish

Academic Editor

PLOS ONE

Journal Requirements:

a) Did participants provide their written or verbal informed consent to participate in this study?

4. Thank you for stating the following in the Funding Section of your manuscript: 

"This work has been supported by German Federal Ministry of Health (BMG) COVID-19 Research and development funding to the World Health Organization"

We note that you have provided funding information. However, funding information should not appear in the Funding section or other areas of your manuscript. We will only publish funding information present in the Funding Statement section of the online submission form. 

"This work has been supported by German Federal Ministry of Health (BMG) COVID-19 Research and development funding to the World Health Organization"

Additional Editor Comments:

Please see the reviewer comments attached.

In addition to those, please specify the timeframe of your study - this will help understand which variant was dominant at the time.

Please also consider including any information on the impact of different VOCs/VOIs in the studied samples.

Reviewers' comments:

Reviewer's Responses to Questions

**Comments to the Author**

1. Is the manuscript technically sound, and do the data support the conclusions?

Reviewer #1: Yes

Reviewer #2: Yes

2. Has the statistical analysis been performed appropriately and rigorously? 

Reviewer #1: Yes

Reviewer #2: Yes

3. Have the authors made all data underlying the findings in their manuscript fully available?

Reviewer #1: No

Reviewer #2: Yes

4. Is the manuscript presented in an intelligible fashion and written in standard English?

Reviewer #1: Yes

Reviewer #2: Yes

5. Review Comments to the Author

Reviewer #1: Thanks for the authors for the good work, comments as below:

1- In introduction section line -75 & discussion section line-318, It would be great if the COVID-19 epidemiological curve in Jordan during the first 19 months of the pandemic can be demonstrated in a figure that reflect count of confirmed covid-19 cases in community and counts of infected HWs in 8 hospitals in parallel with actions taken at the public & hospital level. See an example below:

https://doi.org/10.1001/jama.2020.11160

2- Please indicate what is the status of 8 hospitals in regard to similarities in IPC practices, PPE shortages and designated COVID-19 hospitals.

3- Under study design & participants enrollment:

Line -115, please define a suspected/probable COVID-19 patients or indicate the references in reference section.

4- Under identification of cases subsection: RDT abbreviation stands for what- need to write it as is in the first time.? Also, what do you mean by suspected criteria A or B in line 124 (please clarify).

5- Under control section: How did you manage HWs with prior positive PCR or RDT , did you test them before enrollment especially they can carry the virus with no symptoms, how can you handle it.

Reviewer #2: This study supports the COVID-19 prevention measures adopted in hospitals located in Jordan as advised by the WHO. The results highlights the importance of personal hygiene in combating viral infections as well as in precautions to pandemic. The study is short but has impact in the current situation.

Minor comments

- The standard method of hand washing needs to be mentioned with other PPEs. A short summary of the training provided to HWs needs to be mentioned.

- Whether the HWs were trained for hand washing steps, especially for this study?

- The HWs were wearing gloves as part of their PPEs. How does the investigators implement hand washing in this context? Do they change the gloves very often during their work?

- Authors stated that HWs exposed to COVID-19 patients within 1 meter for more than 15 minutes increased 3-fold adds of infection. As SARS-CoV-2 is a respiratory virus, a more detailed clarification with respect to the face mask need to be described.

6. PLOS authors have the option to publish the peer review history of their article (what does this mean?). If published, this will include your full peer review and any attached files.

Reviewer #1: **Yes: **Faisal Abdoh Alasmari

Reviewer #2: No

---

## [Author Response · Author response to Decision Letter 0]

22 Apr 2022

Dear Editor,

Many thanks for taking into consideration our paper entitled: Infection prevention and control risk factors in health workers infected with SARS-CoV-2 in Jordan: a case control study.

Please find below our feedback on your recommended amendments:

- We have removed the funding section from the manuscript and added it in the online editorial manager system.

- The authors have declared that no competing interests exist

- I have linked and validate my personal ORCID

- Proprietors of the data are the Jordanian Ministry of Health, which can make such data available upon request. The authors of this study can facilitate access to the dataset through the Jordanian Ministry of Health. The researchers cannot make data available directly on behalf of the Jordanian Ministry of Health. Retrieval of data should be sought via the Jordanian Ministry of Health (iprd@moh.gov.jo)

- Reference n 11 was updated with new access link (https://www.who.int/tools/godata)

- Reference n 30 was updated with new access link (https://apps.who.int/iris/handle/10665/342620)

Reviewer #1: Thanks for the authors for the good work, comments as below:

1- In introduction section line -75 & discussion section line-318, It would be great if the COVID-19 epidemiological curve in Jordan during the first 19 months of the pandemic can be demonstrated in a figure that reflect count of confirmed covid-19 cases in community and counts of infected HWs in 8 hospitals in parallel with actions taken at the public & hospital level. See an example below:

https://doi.org/10.1001/jama.2020.11160

- We acknowledge the important contribution that the suggested graph would add’ however, we have no comprehensive time-data around infections in healthcare workers 

2- Please indicate what is the status of 8 hospitals in regard to similarities in IPC practices, PPE shortages and designated COVID-19 hospitals.

- Thanks for the important input and we added as follows: “At the time of the study, all selected facilities were designated as COVID-19 hospitals, with adequate stock in personal protective equipment, and IPC practices were aligned under the standard protocol devised the Ministry of Health of Jordan with the support of the WHO Jordan Country Office. Specifically, training on the use of personal protective equipment (PPE), including hand washing steps and its complementarity with the use of gloves (remove gloves and proceed to hand washing between patients or between contact with various sites on a single patient), had been ongoing in line with international standards and with practical sessions.”

3- Under study design & participants enrollment:

Line -115, please define a suspected/probable COVID-19 patients or indicate the references in reference section.

- As suggested, we indicated the appropriate reference

4- Under identification of cases subsection: RDT abbreviation stands for what- need to write it as is in the first time.? Also, what do you mean by suspected criteria A or B in line 124 (please clarify).

- We agree on the importance of spelling out and clarified RDT being rapid diagnostic test. We also added the previous reference to clarify that we are referring to the WHO probable and suspected criteria

5- Under control section: How did you manage HWs with prior positive PCR or RDT , did you test them before enrollment especially they can carry the virus with no symptoms, how can you handle it.

- We better clarified this aspect and highlighted the fact that controls were tested before enrollment. The sentence now reads: “Enrolment was preceded by testing and having a positive serology test to SARS-CoV-2 and/or prior vaccination was considered as exclusion criteria”

Reviewer #2: This study supports the COVID-19 prevention measures adopted in hospitals located in Jordan as advised by the WHO. The results highlights the importance of personal hygiene in combating viral infections as well as in precautions to pandemic. The study is short but has impact in the current situation.

Minor comments

1. The standard method of hand washing needs to be mentioned with other PPEs. A short summary of the training provided to HWs needs to be mentioned.

- We clarified and added as follows: “At the time of the study, all selected facilities were designated as COVID-19 hospitals, with adequate stock in personal protective equipment, and IPC practices were aligned under the standard protocol devised the Ministry of Health of Jordan with the support of the WHO Jordan Country Office. Specifically, training on the use of personal protective equipment (PPE), including hand washing steps and its complementarity with the use of gloves (remove gloves and proceed to hand washing between patients or between contact with various sites on a single patient), had been ongoing in line with international standards and with practical sessions.”

2. Whether the HWs were trained for hand washing steps, especially for this study?

- Please see above

3. The HWs were wearing gloves as part of their PPEs. How does the investigators implement hand washing in this context? Do they change the gloves very often during their work?

- Please see above

4. Authors stated that HWs exposed to COVID-19 patients within 1 meter for more than 15 minutes increased 3-fold adds of infection. As SARS-CoV-2 is a respiratory virus, a more detailed clarification with respect to the face mask need to be described.

We expanded on this concept and added: “While our findings on physical distancing of at least 1 metre confirms it as a key IPC and public health and social measure to reduce transmission of SARS-CoV-2, the available evidence mostly stems from studies with small sample sizes and other methodological concerns (22,23,24). On the other hand, this confirms the importance of physical distancing as part of a comprehensive strategy of measures to suppress transmission, including mask wearing (use of masks alone not sufficient to provide an adequate level of protection against COVID-19).”

---

## [Editor Report · Decision Letter 1]

1 May 2022

PONE-D-22-04948R1Infection prevention and control risk factors in health workers infected with SARS-CoV-2 in Jordan: a case control studyPLOS ONE

Dear Dr. Bellizzi,

Thank you for submitting your manuscript to PLOS ONE. After careful consideration, we feel that it has merit but does not fully meet PLOS ONE’s publication criteria as it currently stands. Therefore, we invite you to submit a revised version of the manuscript that addresses the points raised during the review process.

As stated in the previous communication, please clarify the study period and any information on VOC/VOIs on the impact of your findings.

We look forward to receiving your revised manuscript.

Kind regards,

Balram Rathish

Academic Editor

PLOS ONE

Journal Requirements:

Additional Editor Comments:

As stated in the previous communication, please clarify the study period and any information on VOC/VOIs on the impact of your findings.
---

## [Author Response · Author response to Decision Letter 1]

11 May 2022

Editor comments:

As stated in the previous communication, please clarify the study period and any information on VOC/VOIs on the impact of your findings.

Dear Editor, thanks for your inputs and please find below responses to your queries:

- We added in the abstract the following statement (lines 40-41): The study lasted approximately two months (from early January to early March 2021). 

- A very similar statement was added in the Introduction (lines 97-98)

- We added the following statement in the Discussion (lines 333-336): Finally, generalization is not possible due to important differences in the pandemic evolving differently across countries, regions, and hospitals. Specifically, during the study period health workers were most likely affected by the original coronavirus strain, which was less contagious than the subsequent VOCs.

---

## [Editor Report · Decision Letter 2]

24 Jun 2022

Infection prevention and control risk factors in health workers infected with SARS-CoV-2 in Jordan: a case control study

PONE-D-22-04948R2

Dear Dr. Bellizzi,

We’re pleased to inform you that your manuscript has been judged scientifically suitable for publication and will be formally accepted for publication once it meets all outstanding technical requirements.

Kind regards,

Balram Rathish

Academic Editor

PLOS ONE

---

## [Editor Report · Acceptance letter]

28 Jun 2022

PONE-D-22-04948R2 

Infection prevention and control risk factors in health workers infected with SARS-CoV-2 in Jordan: a case control study 

Dear Dr. Bellizzi:

I'm pleased to inform you that your manuscript has been deemed suitable for publication in PLOS ONE. Congratulations! Your manuscript is now with our production department. 

Kind regards, 

on behalf of

Dr. Balram Rathish 

Academic Editor

PLOS ONE